# Remote Sensing-Based Hydro-Extremes Assessment Techniques for Small Area Case Study (The Case Study of Poland)

Monika Birylo * and Zofia Rzepecka

Faculty of Geoengineering, University of Warmia and Mazury in Olsztyn, Oczapowskiego St. 2,
10-719 Olsztyn, Poland
* Correspondence: monika.sienkiewicz@uwm.edu.pl

**Abstract:** Meteorological hazards, such as floods, can develop rapidly and are usually a local phenomenon. On the other hand, drought is a phenomenon arising over longer periods. Moreover, it occurs in areas that are remarkably diverse in terms of surface area. Drought has a massive impact on agriculture, socio-economic activities, and the natural environment. However, determining the losses associated with the phenomenon of drought and its identification is complicated. The aim of this paper is to identify and quantify droughts using climatic indices, which include the combined climatologic deviation index, groundwater drought index, water storage deficit index and multivariate standardized drought index. Based on the research, it was concluded that the CCDI, GGDI, WSDI, and MSDI indicators can be a useful tool, on the basis of which it was possible to analyze drought periods. These periods were not related to changes and loss of groundwater, but resulted from low rainfall and snowfall.

**Keywords:** extremes; GRACE; groundwater; drought; categorization





## 1. Introduction

Droughts can be described as a phenomenon characterized by reduced natural water availability. The combination of variation in the water balance due to drought and natural activities (which may include climatic conditions, subsoil features with specific porosity and permeability), as well as human activity, describes the circulation and abstraction of water [1,2]. Due to the direct influence of droughts on people and businesses there is a need for the continuous assessment, observation, and prevention of this phenomena, as it can cause extreme weather phenomena all over the planet. Drought can occur in areas with high and low rainfall because it results from the balance between precipitation and evapotranspiration. Drought intensity is a relative factor. It seems to depend on the duration, intensity, and scope of the drought episode. However, requirements would also need to be taken into account caused by human activity and vegetation. Even short-term droughts can impact society for many years [3]. Long-term droughts may cause land subsidence, which poses a threat to the stability of the ground, foundations, and streets. Huge damage can also be seen in ecosystems. Some areas may even become unusable, and some will have to completely change crops to less demanding plants. Groundwater shortages can lead even to a lack of drinking water, which leads to local governments' decisions to limit water consumption to watering lawns or irrigating home crops. And such decisions lead to smaller crops and an increase in their costs.

In previous studies, the topic of drought index analysis has already been discussed. In [4], drought characteristics in India were assessed on a catchment scale with the use of TWS (total water storage) changes in the temporal-spatial reference. Observations from the GRACE mission and rainfall data were used to characterize drought spread. The combined CCDI (combined climatologic deviation index) and GRACE-DSI (gravity recovery and climate experiment-drought severity index) were modeled. The GRACE-DSI was found to show significant negative trends in most Indian watersheds when compared to the

CCDI. Based on the analyses, it was found that most of the droughts in India are caused by the depletion of TWS. The difference between CCDI and GRACE-DSI is that CCDI uses precipitation and TWSA change observations while GRACE-DSI uses only TWSA (total water storage anomaly) change data [4].

Africa has been repeatedly hit by the disaster of drought in recent decades. This caused significant damage to the environment, both social and economic. Therefore, drought monitoring in many countries of the continent becomes important. In [5], a CCDI (combined climatologic deviation index) index trend was specified on the African continent [5].

In Europe, approximately 65% of drinking water comes from the groundwater. The level of these waters falling below average will become a huge threat to water security. This state is caused by seasonal, multi-seasonal, or even long-term episodes of meteorological drought. Meteorological drought is caused by water runoff spreading through the river basin into the groundwater system. The drought episodes of 2010–2012, 2015, and 2017–2018 showed high spatial coherence over large European areas. Such a situation therefore requires cross-border monitoring and an analysis of groundwater level fluctuations. There was no such joint initiative before, which could have made water management rational. The European Groundwater Drought Initiative was established for research on GDI (groundwater drought index). This provides an assessment of the spatiotemporal changes in the state of the groundwater drought.

Groundwater complements surface flows, as well as the ecosystems in the immediate vicinity in the initial stages of drought. However, such replenishment takes much longer when the drought begins to subside. In [1], the standardized groundwater level index (SGI) was developed and analyzed, which can be used to identify and assess the phenomenon of drought in the Baltic area (Lithuania, Latvia, Estonia). It was noted that in the case of a groundwater drought signal, there is usually a delay and significant attenuation compared to meteorological drought. This is due to the influence of land cover, soil properties, hydraulic properties of both saturated and unsaturated zones, and surface runoff.

Yu et al., 2019, assessed the drought conditions in Mongolia using GRACE observations in the years 2002–2017 by determining the water storage deficit (WSD), which is used to identify dries and to calculate the water scarcity index (WSDI). In the second stage of the work, the WSDI was compared with the standardized precipitation index (SPI) and the standardized precipitation evapotranspiration index (SPEI). Based on the research, two key points of the WSD in 2007 and 2012 were identified [6].

The WSDI was used by [7] to quantify the response to meteorological drought. The impact of global drought on water storage deficit was estimated. Using the Emergency Events Database (EM-DAT), it was confirmed that over 90% of global droughts since 2002 to 2019 led to a water storage deficit. It was found that the water storage deficit was caused by the more severe drought. It has been found that periods shorter than 9 months can cause a storage deficit in low-latitude regions. In high latitude regions, the time scale is longer.

The aim of this paper was to evaluate drought indices using remote sensing-based data. In the light of the currently rapidly changing climate and the related effects, such as huge droughts in some regions, the shortage of drinking water, lack of precipitation and rising average temperature, and on the other hand, floods and excess river waters, the constant monitoring of these indices becomes necessary. This is important due to the fact that drought has a key impact on economy, especially agriculture, economic security, as well as human existence. For monitoring purposes, it becomes extremely useful to apply analysis of the climate indices, such as combined climatologic deviation index (CCDI), groundwater drought index (GDI), water storage deficit index (WSDI) and multivariate standardized drought index (MSDI). Central Europe, as it is an area where extreme phenomena do not occur (like catastrophic droughts occurring sometimes in southern Europe, hurricanes, monsoons, earthquakes) is not of great interest in scientific publications. However, this does not mean that such areas should not be explored. In the light of changing climate, each area should be monitored regularly. This work, based on the example of Poland and

its catchment area, attempts to answer the questions of whether this part of Europe is at risk of drought, how quickly the climate change is progressing and what the causes of water conditions changes are. Similar topics regarding groundwater changes in this region have been previously discussed among others in publications [8], but only for the Warsaw urban area, where one piezometer and two wells were analyzed; [9] for the area of Lebiedzianka river basin; [10] and for the same area but without taking into account meteorological indicators. The topic of meteorological indices was discussed in [11] but concerning UTCI, STI, Oh_H, WL, and OV indices; in [12,13] the standardized precipitation index was used.

## 2. Data and Case Study Localization

The performed analyses concern the territory of Poland. This country covers an area of approximately 312,000 km$^2$, extending between the parallels 49° and 55° of the northern latitude and between the meridians 140 to 240 of the eastern longitude (Figure 1). The Polish climate is generally temperate, with a strongly marked seasonality. The western part of Poland is slightly more influenced by the oceanic climate, while the eastern part is more affected by the continental weather. In the southern part, where there are mountain regions, the climate is slightly colder with a higher precipitation than in the northern part of the country. Two river basins cover almost the entire area of Poland, these are the Vistula basin (with an area of about 194,500 km$^2$) and Odra basin (covering an area of about 118,900 km$^2$). According to the soil map of Poland (distributed by the Soil Science and Agricultural Chemistry Committee of the Polish Academy of Science and the National Soil Science Association in the scale of 1:600,000), the brown soils are the most common; the other soils we encounter are podzol, organic, alluvial, black, and rendzinas soils.

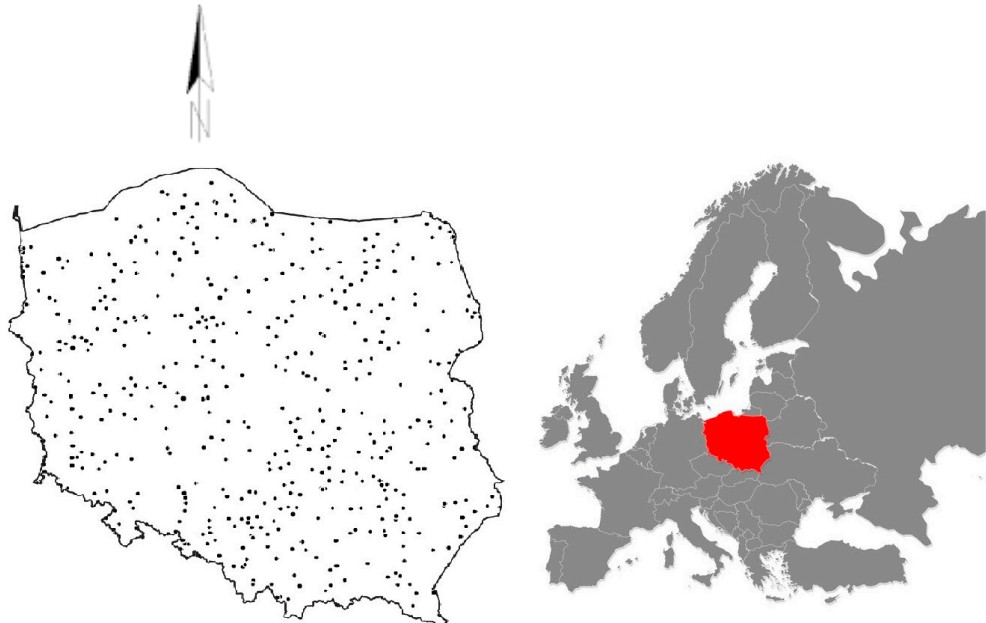

**Figure 1.** Location of the case study area (Poland) along with the location of its wells (black dots). The length of Poland is 14–24°E, 49–54°N.

The mean porosity coefficients are on the level of 0.41 and 0.42 for the Vistula basin and Odra basin, respectively [10]. These values can be used to scale between the groundwater level (GWL) and groundwater storage (GWS).

The method of recomputing the GWL into GWS was introduced by an author. The determination of the porosity coefficient is calculated at the location of each of the measuring wells. It consists of using the analysis of soil profiles, origin, soil structure, and above all, the permeability of each of the profile elements, and on this basis, the flow of water into the groundwater table is averaged. The scheme is presented in Figure 2. The results of the soil

approach to weighting the GWL values were confirmed by the statistical method described in [14].

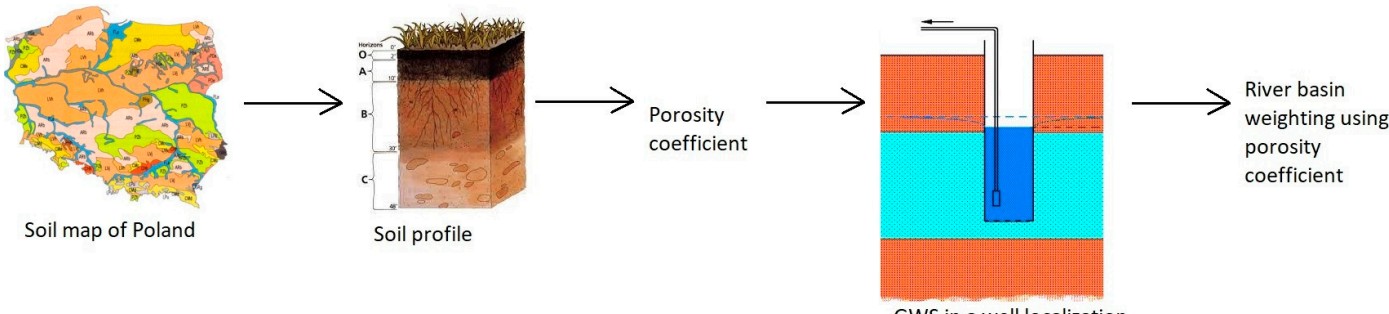

**Figure 2.** Flowchart—porosity coefficient computation using soil approach.

In this paper, the data acquired from the GRACE mission and MERRA2 system for the territory of Poland are used.

The GRACE (Gravity Recovery and Climate Experiment) mission purpose was to track the Earth's gravity field changes. The mission consists of two similar satellites placed in near-circular orbits at ~500 km altitude and ~89.5° inclination, separated from each other by approximately 220 km and linked by a highly accurate inter-satellite microwave ranging system. GRACE is a modern tool measuring water mass shifts connected with changing seasons, weather, and climate processes. After proper processing, the GRACE observations reflect the total water storage (TWS) variations. The processed data can be acquired from the three main computational centers, i.e., GFZ (GeoforschungsZentrum, Potsdam), CSR (Center for Space Research at University of Texas, Austin), and JPL (Jet Propulsion Laboratory) and in two forms: spherical harmonics coefficients and mascons. The name "mascon" is an abbreviation of mass concentration blocks. For the purpose of this article, the JPL mascon solution, provided with a resolution of 0.5° was chosen [15]. This approach gave the GRACE TWS results for ten hydrological years, from November 2002 to October 2022 [16].

The development of the MERRA 2 model (The Modern-Era Retrospective Analysis for Research and Applications, Version 2) is based on the prediction of different observation combinations. The result of the data assimilation is a GRID (0.5° lat × 0.625° lon × 72 hybrid levels) for a wide range of variables. It is important that it includes variables that are not observed directly. The atmospheric reanalysis using the latest satellites is carried out by NASA's Global Modeling and Assimilation Office (GMAO). The model update and analysis takes place in the Goddard Earth Observing System (GEOS) [17].

The GRACE and MERRA satellite observations provide a suitable tool for the quick and cheap monitoring of water conditions, an aspect that is both a disadvantage and an advantage—the existence of an unprecedented possibility of global surveys and surveys of large areas; however, this resolution limits the possibilities of assessing small areas and makes point surveys impossible. The GRACE and MERRA observation tiles were carried out with the same spatial division: 16 tiles for the Odra basin and 25 tiles for the Vistula basin, as presented in [10].

The groundwater level (GWL) data were obtained from the Polish Hydrogeological Annual Reports, from 2002 to 2022. Among the many observed wells, 69 wells were selected, which were continuously measured throughout the whole period of November 2006 to October 2022. The measured and reported variation depths were recomputed into appropriate GWL changes by inverting the sign of the change and applying a porosity coefficient. Averaging over the Vistula and Odra basin areas gave more uniform results, which are believed to reflect the average behavior of the amount of groundwater in Poland [18–33].

## 3. Methods

The paper presents an attempt at adopting geodetical remote-based observations from the GRACE mission in the form of total water storage changes supported by an assimilation model and in situ data to evaluate and monitor climate change. For this purpose, indices like the combined climatologic deviation index (CCDI), groundwater drought index (GDI), water storage deficit index (WSDI), and multivariate standardized drought index (MSDI) were adopted. The use of data and its re-computation into indices are presented in the form of a flowchart (Figure 3).

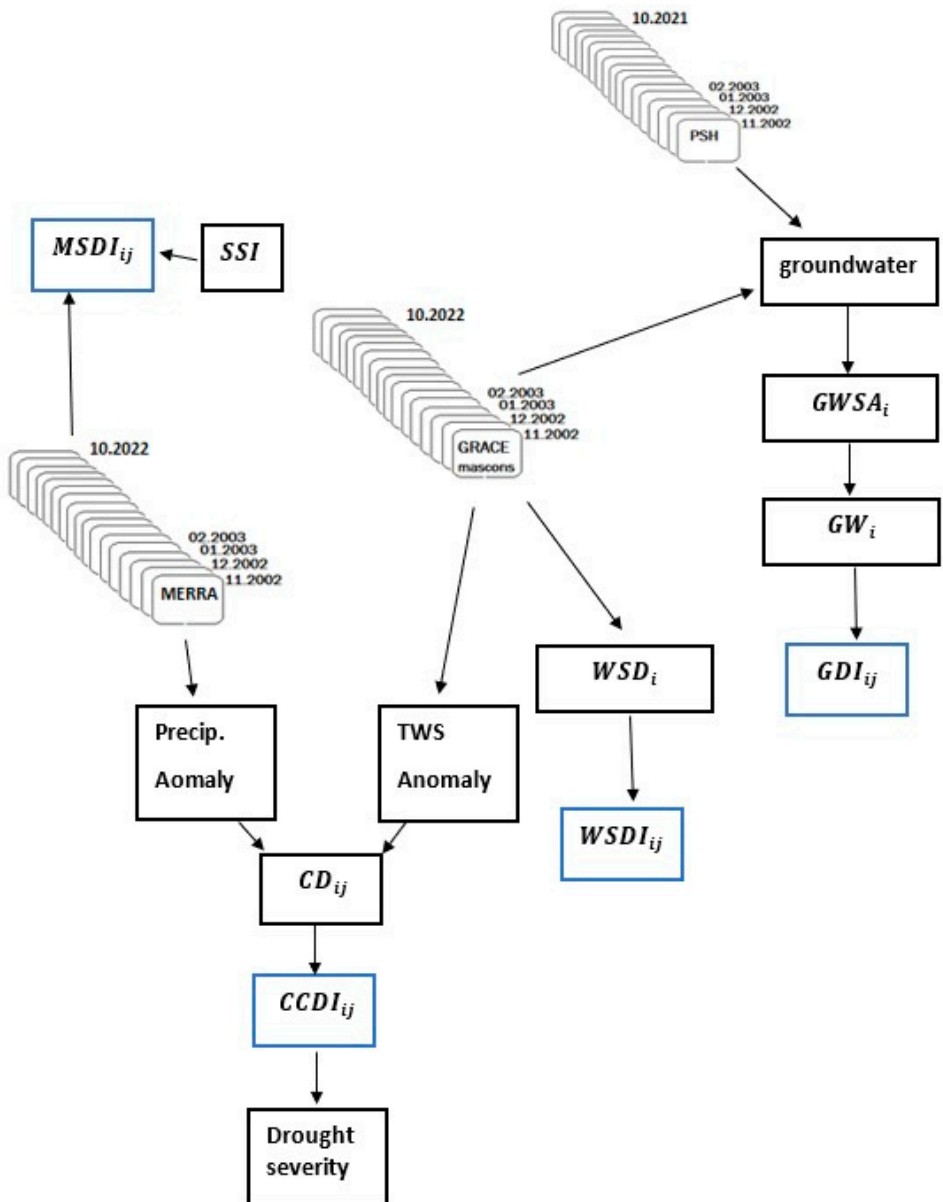

**Figure 3.** Flowchart—data flow and final indices computed in this paper.

### 3.1. Combined Climatologic Deviation Index

The combined climatologic deviation index (CCDI) is a combination of atmospheric and terrestrial water. The idea of the index is to take into consideration meteorological-, hydrological-, agricultural-, and human-influenced drought occurrences [34]. The CCDA is formulated as a sum of the total water storage anomalies (TWSA) monthly observations and the monthly precipitation anomaly (PA) [34]:

$$PA_{i,j} = P_{i,j} - \overline{P} \text{ [cm]}, \tag{1}$$

where $PA_{i,j}$—precipitation anomaly in year $i$, month $j$, $P_{i,j}$—amount of precipitation in a year $i$ and month $j$, $\overline{P}$—monthly average precipitation.

The residuals can be computed as [34]:

$$PA_{i,j}^{R} = PA_{i,j} - \overline{PA_j} \text{ [cm]} \tag{2}$$

where $PA_{i,j}^{R}$—precipitation anomaly residual in a year $i$ and month $j$, $\overline{PA_j}$—average precipitation anomaly in month $j$.

Essential for computing the CCDI is defining the total water storage anomaly residuals based on the GRACE observations [34]:

$$TWSA_{i,j}^{R} = TWSA_{i,j} - \overline{TWSA_j} \text{[cm]} \tag{3}$$

where $TWSA_{i,j}^{R}$—total water storage anomaly residual in a year $i$ and month $j$, $TWSA_{i,j}$—total water storage anomaly in a year $i$ and month $j$, $\overline{TWSA_j}$—monthly average total water storage anomaly in a month $j$.

Having previously computed the precipitation anomaly residuals and total water storage anomaly residuals, the next step is to compute the combined precipitation and total water storage anomaly deviations for a particular year and month [34]:

$$CD_{i,j} = PA_{i,j}^{R} + TWSA_{i,j}^{R} \text{[cm]} \tag{4}$$

Now it is possible to compute the combined climatologic deviation [34]:

$$CCDI_{i,j} = \frac{(CD_{i,j} - \overline{CD})}{st.dev.(CD)} \text{[cm]} \tag{5}$$

where $CCDI_{i,j}$—combined climatologic deviation index, $CD_{i,j}$—combined precipitation and total water storage anomaly deviations, $\overline{CD}$—monthly average value of combined precipitation and total water storage anomaly, $std(CD)$—standard deviation of combined precipitation and total water storage anomaly.

### 3.2. Drought Severity

Based on the computed combined climatologic deviation index, the drought severity index was computed [34]:

$$DS = \sum_{n}^{m} CCDI \, (m - n + 1) \tag{6}$$

Knowing the value of the CCDI, the drought severity can be determined according to Table 1.

**Table 1.** Categories of drought severity [35,36].

| CCDI [cm] | WSDI [cm] | Category of DS with Severity Level |
|---|---|---|
| $[-1.45, -\infty)$ | $[-3, -\infty)$ | Extreme drought (D4) |
| $[-1.44, -0.94]$ | $[-3, -2]$ | Severe drought (D3) |
| $[-0.93, -0.46]$ | $[-2, -1]$ | Moderate drought (D2) |
| $[-0.45, -0.28]$ | $[-1, -0]$ | Mild drought (D1) |
| $[0.28, -0.44]$ | $[-1, 1]$ | Normal (No) |
| $[0.45, 0.28]$ | $[0.5, 1]$ | Mild wet (W1) |
| $[0.93, 0.46]$ | $[1, 1.5]$ | Moderate wet (W2) |
| $[1.44, 0.94]$ | $[1.5, 2]$ | Severe wet (W3) |
| $(\infty, 1.45]$ | $(\infty, 2]$ | Extreme wet (W4) |

The water storage anomaly was computed by comparing the monthly data to a mean long-term time series values. A way of assessment of the groundwater, especially taking into account drought characteristics, is the groundwater drought index (GGDI). It can be computed based on a monthly climatology ($GW_i$) [37]:

$$GW_i = \frac{1}{n_i}\sum_{1}^{n_i} GWSA_i \tag{7}$$

where $i$—1, 2, 3, 4, 5, 6, 7, 8, 9, 10, 11, 12, $GWSA_i$—groundwater storage anomaly for month $i$ and year $j$, $n_i$—month number.

And so, the groundwater drought index can be computed by normalizing the groundwater storage deviation [38]:

$$GGDI_{i,j} = \frac{\left(GSD_{i,j} - \overline{GSD}\right)}{st.dev.(GSD)} \tag{8}$$

where $GGDI_{i,j}$—groundwater drought index, $\overline{GSD}$—mean groundwater storage deviation, $std(GSD)$—standard deviation of groundwater storage deviation.

*3.3. Water Storage Deficit*

Differencing the time series of the GRACE total water storage anomaly and the average monthly total water storage anomaly value is the water storage deficit [39]:

$$WSD_{i,j} = TWSA_{i,j} - \overline{TWSA_j} \tag{9}$$

where $WSD_{i,j}$—water storage deficit, $TWSA_{i,j}$—total water storage anomaly, $\overline{TWSA_j}$—long-term mean value of total water storage anomaly.

When introducing the mean normalization method and comparison of the water storage deficit with mean and standard deviation of WSD, we have [39,40]:

$$WSDI = \frac{WSD - \mu}{\sigma} \tag{10}$$

where $WSDI$—water storage deficit index, $\mu$—mean value of water storage deficit, $\sigma$—standard deviation of water storage deficit.

Knowing the value of WSDI, the drought severity can be determined according to Table 1.

*3.4. Multivariate Standardized Drought Index*

The essential components of agricultural and meteorological conditions assessment in terms of drought monitoring are the precipitation and soil moisture [41]. The drought conditions can be based on the computation of multivariate standardized drought [41]:

$$MSD = \varphi^{-1}(P) \tag{11}$$

where $\varphi$—normal distribution composed of joint probability of soil moisture and precipitation.

Taking into account the accumulated precipitation (in the form of standardized precipitation index—SPI) and accumulated soil moisture (standardized soil moisture index—SSI), a multivariate standardized drought index can be determined [35]:

$$MSDI^{(1)} = P\left(AP \leq AP_{n+1,m}^{(1)}, \ A \leq AS_{n+1,m}^{(1)}\right) \tag{12}$$

where $n \to MSDI^{(1)}, MSDI^{(2)}, MSDI^{(3)}, MSDI^{(4)}, \ldots, MSDI^{(n)}$.

Knowing the value of the MSDI, the drought severity can be determined according to Table 2.

**Table 2.** Categories of MSDI [35,36].

| MSDI [cm] | Category of DS with Severity Level |
|---|---|
| $[-2, -\infty)$ | Exceptional drought (D4) |
| $[-1.6, -1.99]$ | Extreme drought (D3) |
| $[-1.3, -1.59]$ | Severe drought (D2) |
| $[-0.8, -1.29]$ | Moderate drought (D1) |
| $[-0.5, -0.79]$ | Abnormally dry (D0) |
| $[0.5, 0.79]$ | Abnormally wet (W0) |
| $[0.8, 1.29]$ | Moderate wet (W1) |
| $[1.3, 1.59]$ | Severe wet (W2) |
| $[1.6, 1.99]$ | Extreme wet (W3) |
| $(\infty, 2]$ | Exceptional wet (W4) |

## 4. Results

The standardized indices can be a very helpful tool in hydrological and climatic research. Their advantage is that they combine plenty of variables that have an influence on climatic changes. The research was conducted in the area of the Vistula basin and the Odra basin (Figure 1). The same division was taken into account according to the well data. The thorough and constant monitoring of water levels carried out through designated indices would allow for a faster response to changing levels and prevent or remedy droughts or floods. Such an analysis will also allow for better planning in agriculture, water management, and rational use of drinking groundwater.

### 4.1. Combined Climatologic Deviation Index

Firstly, the combined climatologic deviation index was computed, based on the TWS changes values from the GRACE mission observations (Figure A1) and the total precipitation values from the MERRA2 (Figure A2) assimilation model (according to the flowchart—Figure 3) using Formula (5).

The analysis of the time series presented in Figure 4 shows stable and seasonal amplitude changes between W2 and D4 till the end of 2009; from the beginning of 2010 a 4-year period of a very dry condition of the CCDI occurs; from 2015, the CCDI differences vary from D4 to W4 in the seasonal manner. The Vistula basin CCDI time series is 0.5 cm lower than the Odra basin values, taking into account the GRACE observations. The CCDI computed based on the GRACE FO observations is characterized by a two-month time lag between the Odra and Vistula basins. The lag shows that in recent years, in the Odra basin area, the maximum and minimum CCDI levels are reached two months earlier than in the case of the Vistula River. This proves a higher ratio of precipitation to evaporation in the Odra basin area. This is understandable, because the western part of Poland has a warmer climate and a much earlier spring, as well as a shorter winter.

The basic statistic characteristics of CCDI are presented in Table 3, both for the Vistula and Odra basins. The difference in the maximum values is particularly noticeable.

**Table 3.** Basic statistic characteristics—CCDI.

| Stat. Char. | Vistula Basin [cm] | Odra Basin [cm] |
|---|---|---|
| Max. | 1.836 | 2.412 |
| Min. | −3.935 | −3.720 |
| Mean | −1.081 | −0.810 |
| St. Dev. | 1.002 | 1.002 |

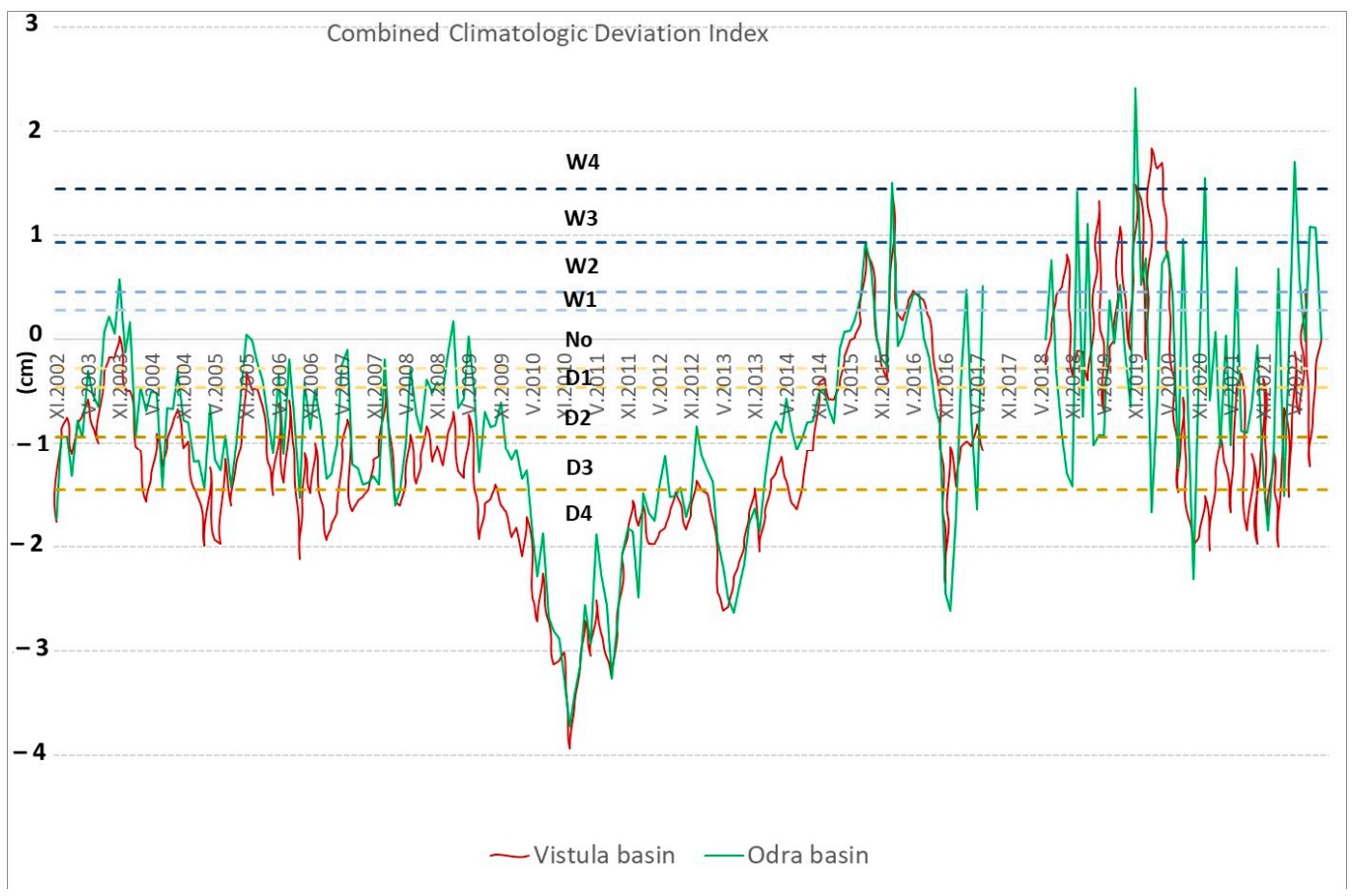

**Figure 4.** Combined climatologic deviation index for Vistula basin and Odra basin together with drought severity categorization.

### 4.2. Groundwater Drought Index

The groundwater storage was previously computed for the researched area of the Vistula and Odra basins [10,14], where the porosity coefficient was calculated based on the assumptions from Figure 2. In the mentioned papers, the GLDAS model was taken into account [2]. Same computations were repeated in this paper but using the MERRA 2 assimilation model.

The analysis of the time series of the groundwater drought index, calculated according to Formula (8), given in Figure 5, showed a higher maxima of the GGDI for the Vistula compared to the Odra basin observed until the summer of 2010 (up to about 0.5 cm). In the next decade, the maximum values of GGDI reached in both basins are similar. A large maximum jump, being at the same time in antiphase to the Odra basin, was noticed in X2020. The minimum values for both basins remain at the same level, except for two periods in which a significant decrease in the Vistula GGDI was observed—III.2013 and III.2016 (by about 1.5 cm). In the case of the Vistula basin, a greater variability of the course of the time series month to month was also noticed—in the periods of X.2006–X.2007 and II.2013–XI.2014.

In the case of the basic statistical characteristics of the GGDI (Table 4), the observed differences in the time series values are confirmed. Significant differences in maximum values were noted.

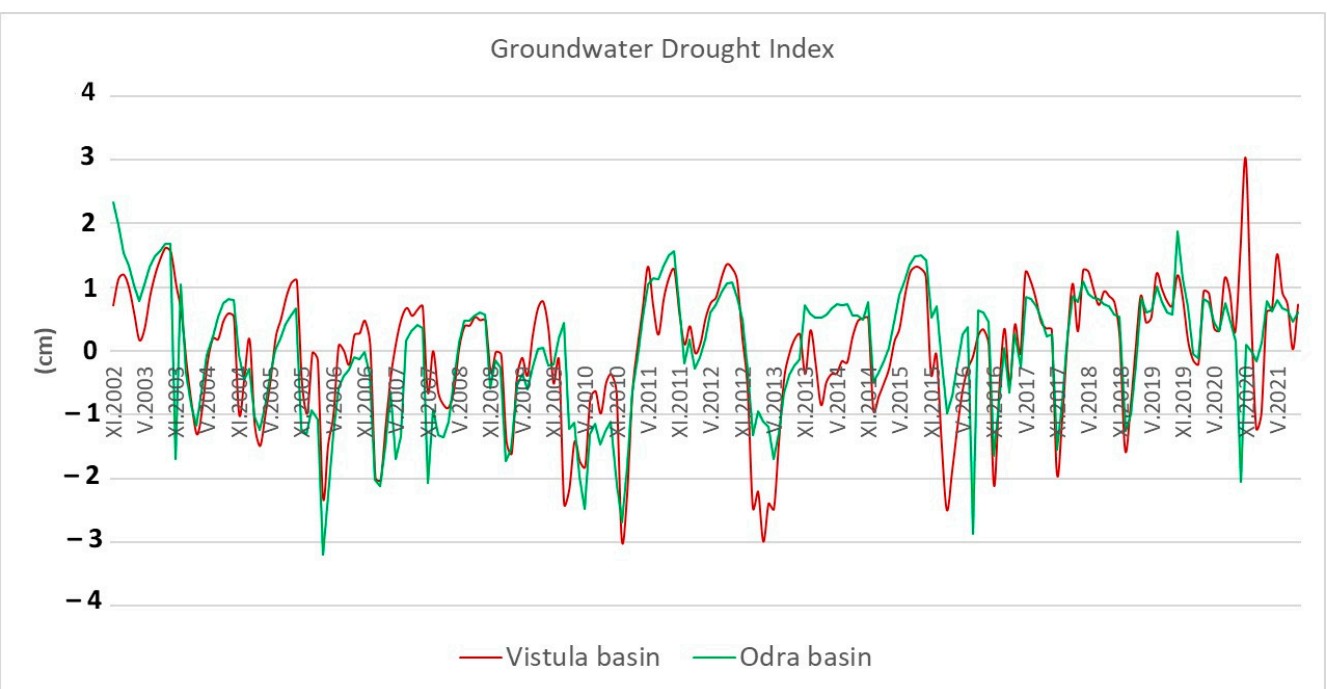

**Figure 5.** Groundwater drought index computed with PSH wells for Vistula and Odra basins.

**Table 4.** Basic statistic characteristics—GDI.

| Stat. Char. | Vistula Basin [cm] | Odra Basin [cm] |
|---|---|---|
| Max. | 3.021 | 2.327 |
| Min. | −2.986 | −3.205 |
| Mean | 0.000 | 0.000 |
| St. Dev. | 1.002 | 1.002 |

*4.3. Water Storage Deficit*

A very important part in groundwater changes monitoring is the permanent observation of the total water storage changes in terms of tracking its surplus and deficit. The water storage deficit index (WSDI), calculated from the monthly changes in the GRACE TWS anomalies, is an excellent tool for identifying the occurrence of drought and characterizing its severity (computed with Formula (10)).

On the basis of the analysis of Figure 6, very similar courses of time series were noticed for both studied basins. Moreover, the course of changes in the WDSI shows a strong seasonal influence, with maximum values in the spring months and minimum values in the autumn months. This is easily explained, because in the study area, spring is characterized by a large amount of precipitation and the dissolution of post-winter snow, while in autumn, there is less evaporation and less precipitation. It is also easy to notice a significant increase in the WSDI value at the turn of 2010 and 2011, when a catastrophic flood took place in Poland. During this period, the maximum values are twice as high (approx. 3 cm) compared to the usual values (approx. 1.5 cm). In this period, the minima are also higher, because they are positive throughout the period. When examining the series from the GRACE-FO mission, the differences in the course of the time series of both catchments have already been found, the graphs are not as well correlated as in the case of the observations from the GRACE mission.

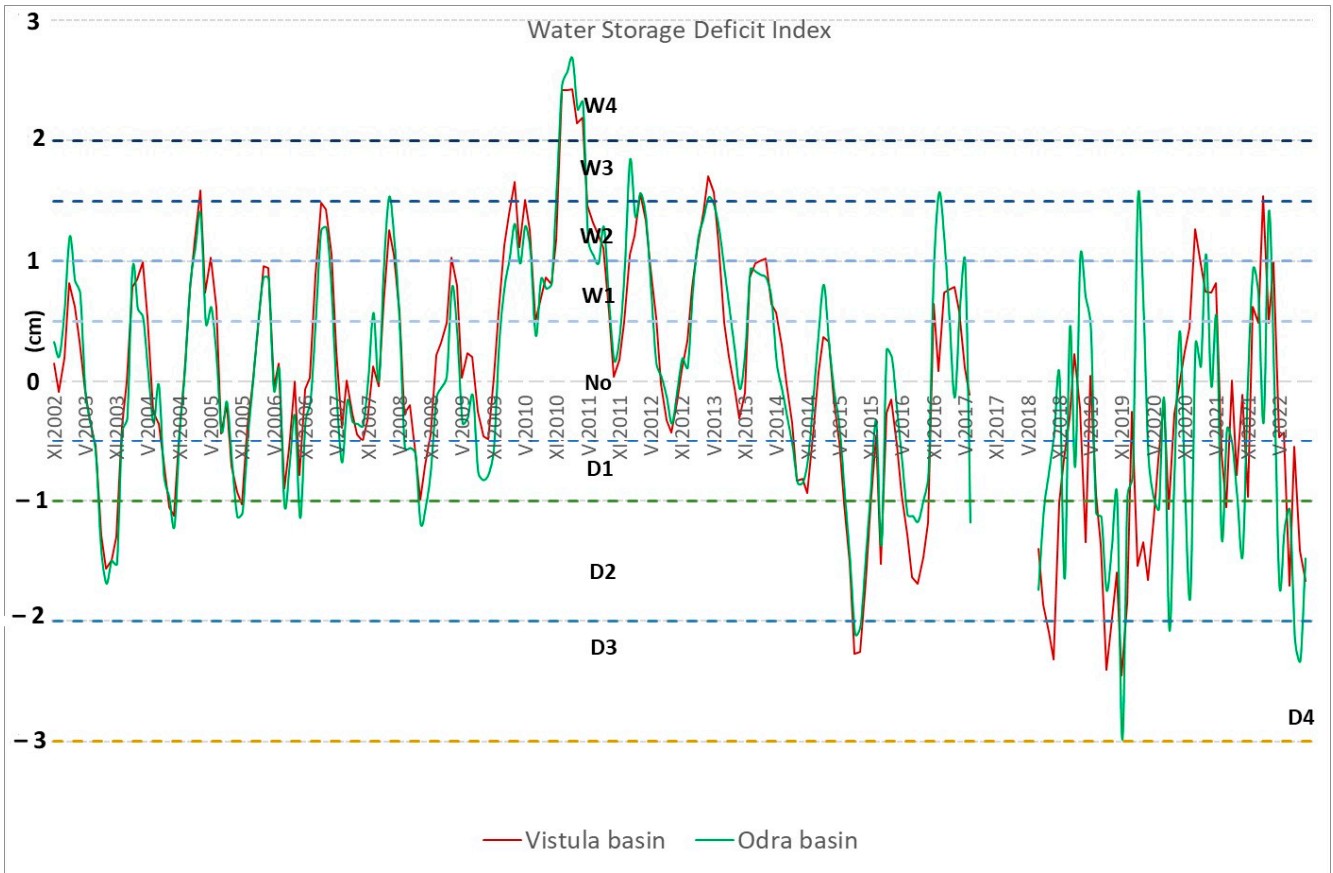

**Figure 6.** Water storage deficit index for Vistula basin and Odra basin.

Outside of the flood season, the WSDI values fall between the W3 and D3 ranges (W3 in the spring months, D3 in the autumn months). During the period of flooding, the minimum values are in the range of W2, during the maximum accumulation they significantly exceed the range of W4.

Based on the analysis of Table 5, it was found that the maximum values are 2.425 cm. Again, the mean WSDI of both catchments is at 0 cm.

**Table 5.** Basic statistic characteristics—WSDI.

| Stat. Char. | Vistula Basin [cm] | Odra Basin [cm] |
|---|---|---|
| Max. | 2.425 | 2.685 |
| Min. | −2.455 | −2.983 |
| Mean | 0.000 | 0.000 |
| St. Dev. | 1.002 | 1.002 |

*4.4. Multivariate Standardized Drought Index*

The last research was carried out taking into account two parameters—meteorological drought (SPI—standardized precipitation index) and agricultural drought (SSI—standardized soil moisture index); both can be computed based on the observables available at: http://drought.eng.uci.edu/ (accessed on 24 September 2023), available in the period XI.2002–X.2016 (using Formula (12)).

Figure 7 shows the SPI, SSI, and MSDI for the Vistula basin area. Higher values of the SPI coefficient are clearly visible, indicating a meteorological drought, especially in the years 2003–2007, when the SSI and MSDI values vary between D3 and D1 (−0.75 cm−−2 cm), and the SPI in the spring period reaches W0 (9.75 cm–1.5 cm). From 2007 to the end of

2014, the SSI and MSDI values are almost equal to the SPI values in the spring period (0.75 cm–1.25 cm). In 2015, the situation is the same as for the first period analyzed. The values of the SSI and MSDI are well-correlated (0.74), but it is worth noting that from 2007, there is a 3-month lag between the MSDI and SSI.

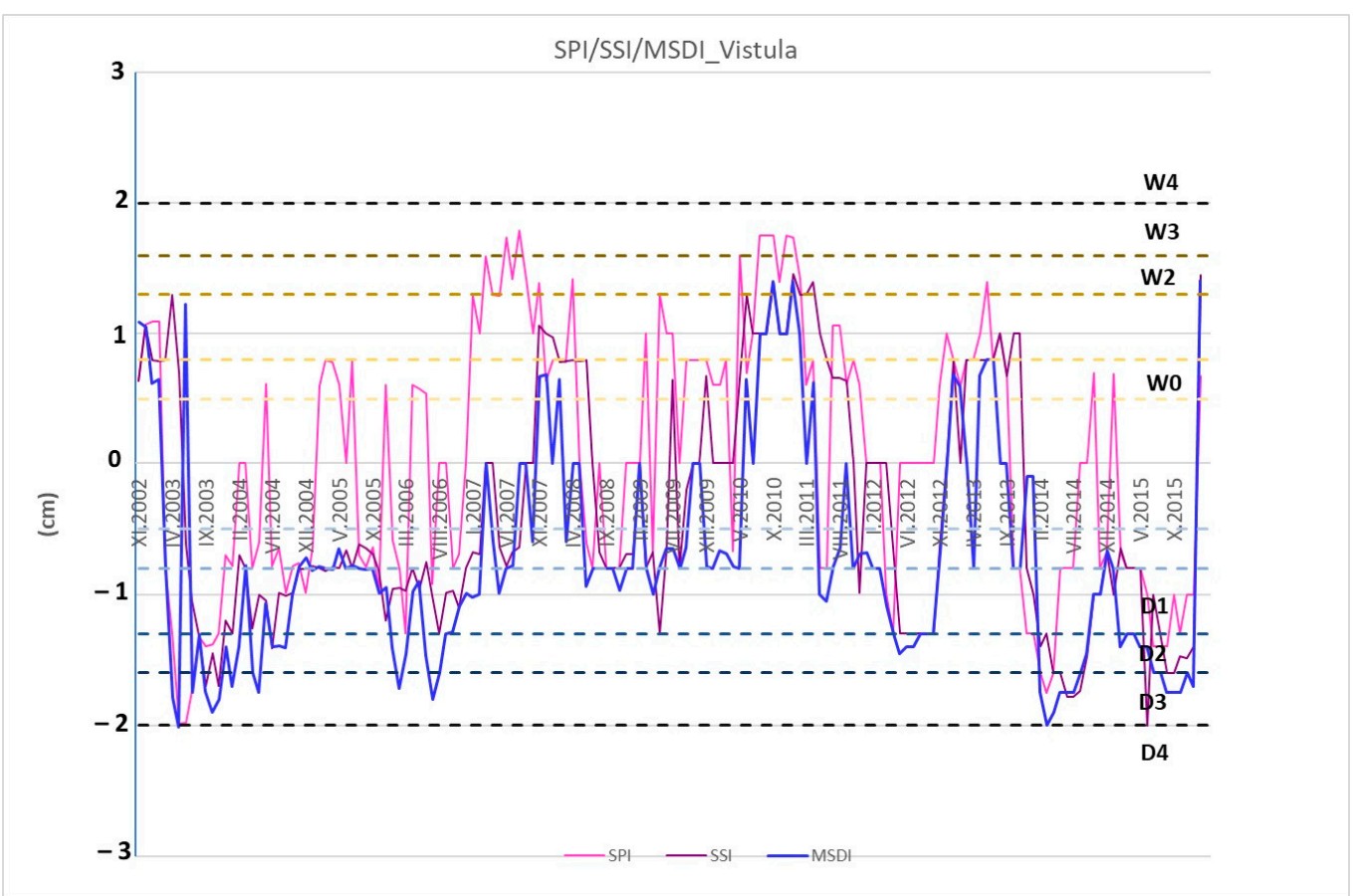

**Figure 7.** Multivariate standardized drought index, SPI/SSI/MSDI for Vistula basin.

Figure 8 shows the time series of the SPI, SSI, and MSDI coefficients of the Odra basin. Based on the time series analysis, the SPI values were found to be higher than the SSI and MSDI throughout the study period—on average by 1 cm in relation to the SSI and almost 2 cm in relation to the MSDI. This steady trend is disrupted in the year of the flood, 2010, when the SPI, SSI, and MSDI values converge to W2/W3 (1.25 cm–1.5 cm). The maximum achieved values were noticed at the turn of spring and summer. No lags were found.

In Table 6, the basic statistical characteristics of the SPI, SSI, and MSDI of both studied basins are presented. Higher maximum values were recorded in the case of the Vistula basin: they are 1.790 cm for the SPI coefficient, 2.000 cm for the SSI coefficient, and 1.750 cm for the MSDI coefficient; while for the Odra basin, 1.460 cm was calculated for the SPI, 1.450 cm for the SSI, and 1.415 cm for the MSDI. It is also possible to notice closer maximum values for all the studied coefficients in the case of Odra. On the other hand, the minimum values are very similar for all the basins' coefficients, around −2 cm. Only the MSDI value for the Odra basin is lower; it amounts to −1.875 cm. It is also interesting to analyze the average values of the time series. The basin of the Vistula is characterized by great diversity, for which the average value of the SPI coefficient is 0.046 cm, SSI 0.098 cm, and MSDI 0.072 cm. On the other hand, for the Odra basin, the averages of all three coefficients are about 0.4 cm.

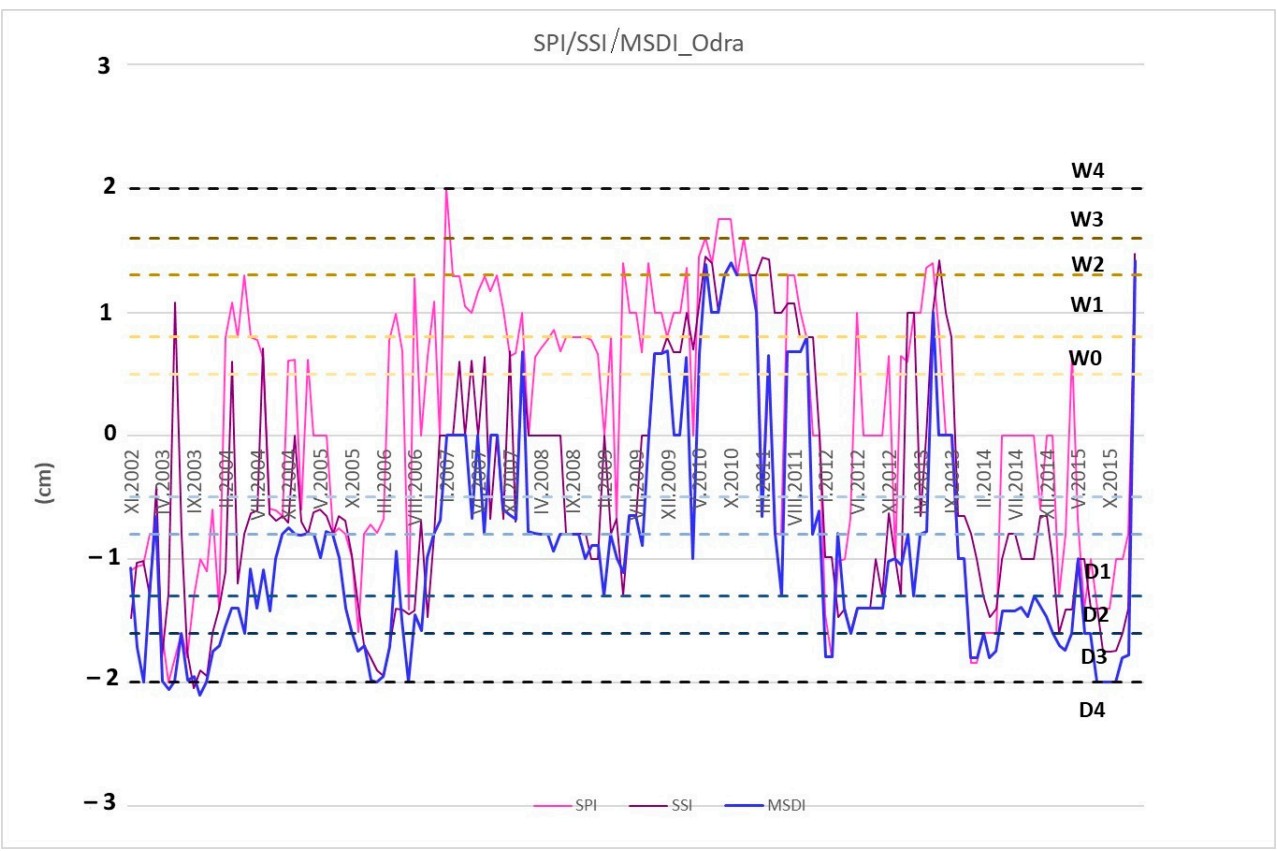

**Figure 8.** Multivariate standardized drought index, SPI/SSI/MSDI for Odra basin.

**Table 6.** Basic statistic characteristics—MSDI.

| Stat. Char. | SPI | SSI | MSDI | SPI | SSI | MSDI |
|---|---|---|---|---|---|---|
| | Vistula Basin [cm] | | | Odra Basin [cm] | | |
| Max. | 1.790 | 2.000 | 1.750 | 1.460 | 1.450 | 1.415 |
| Min. | −1.990 | −2.000 | −1.995 | −2.000 | −2.050 | −1.875 |
| Mean | 0.046 | 0.098 | 0.072 | −0.458 | −0.458 | −0.407 |
| St. Dev. | 0.982 | 1.028 | 0.930 | 0.908 | 0.976 | 0.859 |

## 5. Conclusions

The aim of this paper was to identify and assess droughts using climate indicators such as the combined climatologic deviation index (CCDI), groundwater drought index (GDI), water storage deficit index (WSDI) and multivariate standardized drought index (MSDI). The research was conducted in two basins located in Central Europe: the basins of the Vistula and the Odra. Such catchments were selected due to the greatest availability of the data for the authors. The observations from the GRACE mission, from the MERRA 2 assimilation model, the SPI and SSI determinations, as well as observations of the groundwater level in measuring wells in Poland were used. The research on the mentioned indices is important and has not been discussed before in the areas presented in the publication. As mentioned, the studies were conducted for much smaller areas using direct measurements or using a small range of indices. Based on the conducted research, the following can be concluded:

- In the studied river basins, there were regular periods of drought with an intensity from D1 to above D4. The longest and most intense period of drought, extending over 3 years, is observed for both catchments in the years 2010–2013. During this period, the indicator fluctuated between D1 and below D4, never reaching the W range;
- Much smaller amplitudes of changes between the intervals D and W were observed in the period before the month-long drought of 2002–2010 (−1.5 cm–0.5 cm), after the drought, the amplitudes of the changes increased and reached a range between −2 and 1.5 cm;
- Drought in the catchments, after the analysis of the CCDI coefficient, occurs every year in the autumn and is greater in the catchment of Vistula in comparison to the Odra catchment.
- Using the GGDI coefficient, a stable groundwater level was found throughout the months under study;
- The WSDI analysis showed the deteriorating state of the total water—in the autumn, the values fell to the D2 range and from 2018 they reach D3 and D4. This shows the loss of total water, less precipitation, less water in the atmosphere, and more evaporation and evapotranspiration caused by the increase in temperature. The amount of snowfall in winter is also reduced;
- MSDI should be analyzed depending on climatic zones—Poland is a rather homogenous country in this respect; however, the division into basins is a vertical division, in contrast to the horizontal distribution of climatic zones. When analyzing the effects of meteorological and agricultural drought in the form of the MSDI index, an unfavorable situation in terms of drought was noticed in the study area, especially since 2014, when even the upper MSDI levels are at the D1 level;
- To sum up, the analysis of climate coefficients in terms of researching and identifying the phenomenon of drought using the CCDI, GGDI, WSDI, and MSDI indicators is a necessary tool. The periods of drought can be seen, especially since 2014. This is not groundwater-related drought; it seems to be due to low rainfall and snowfall.
- The proposed methods for determining the water indices can be used in almost any region. And we think it would be worth implementing them in the continuous monitoring of basin areas. Testing the resources and availability of groundwater, which is crucial for consumption, is of exceptional importance. However, the porosity coefficient should not be used in future work in the case of areas covered with ice, because the ice itself has a significant impact on the permeability there and the ice itself could be treated as a rock, which is only an additional, yet important factor.

**Author Contributions:** Conceptualization, M.B.; methodology, M.B.; validation, M.B. and Z.R.; formal analysis, M.B.; investigation, M.B.; resources, M.B.; data curation, M.B.; writing—original draft preparation, M.B.; writing—review and editing, M.B. and Z.R.; visualization, M.B.; supervision, M.B.; project administration, M.B.; funding acquisition, Z.R. All authors have read and agreed to the published version of the manuscript.

**Funding:** This research received no external funding.

**Data Availability Statement:** Data is found through the author.

**Conflicts of Interest:** The authors declare no conflict of interest.

**Appendix A**

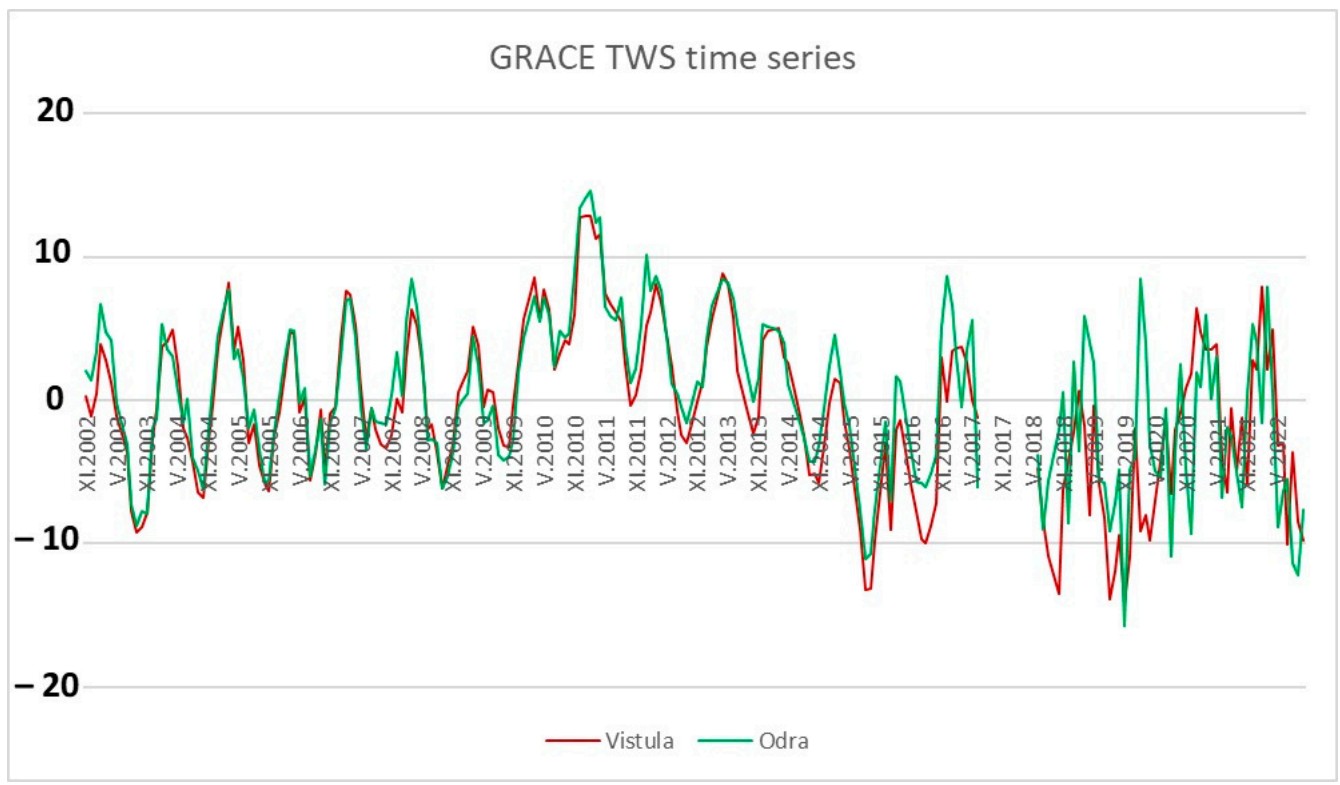

**Figure A1.** GRACE total water storage changes time series for Vistula and Odra basins (in cm).

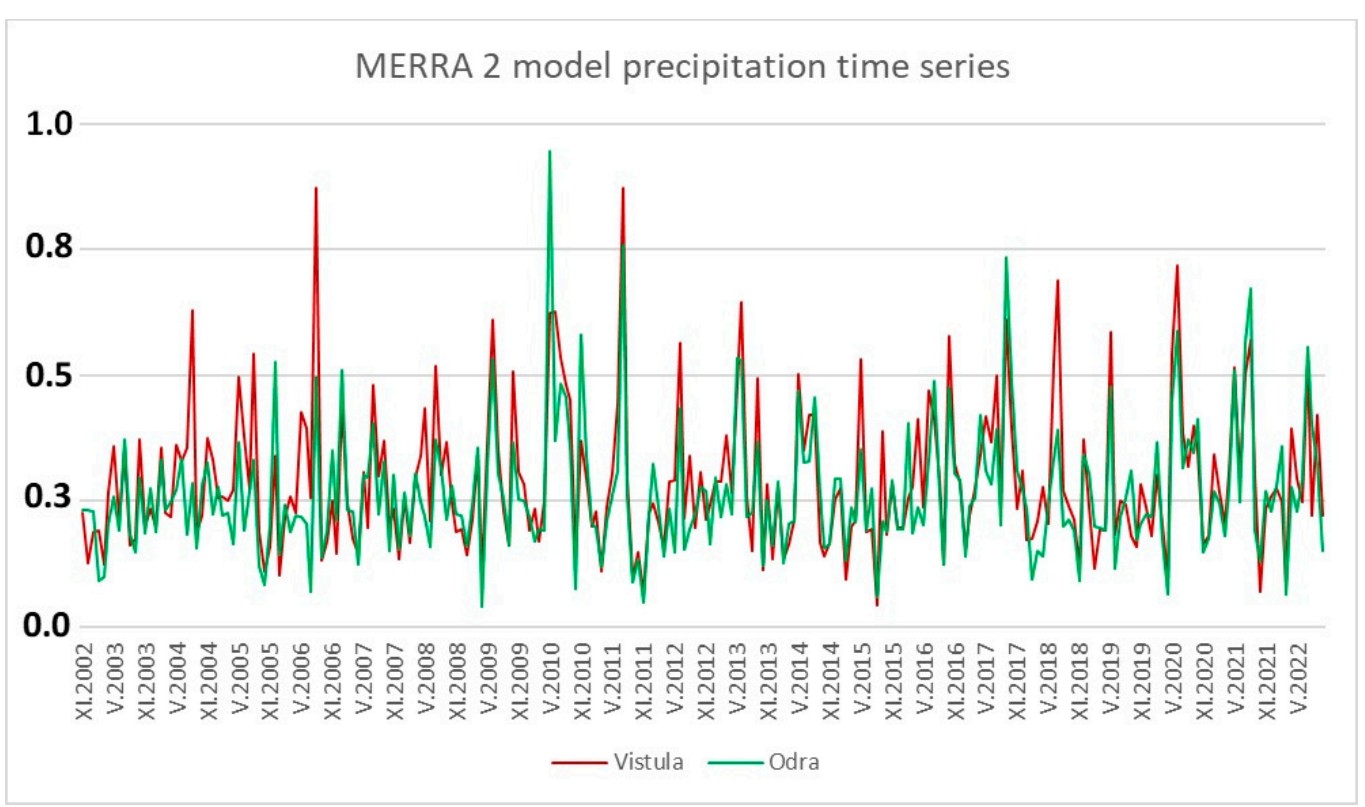

**Figure A2.** Precipitation time series for Vistula and Odra basins from MERRA 2 model (in cm).

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
