# Peer review of "Remote Sensing-Based Hydro-Extremes Assessment Techniques for Small Area Case Study (The Case Study of Poland)"

_remotesensing, doi:10.3390/rs15215226_

Round 1

Reviewer 1 Report

Comments and Suggestions for Authors

The article deals with a very interesting topic regarding the ‘Remote sensing based hydro-extremes assessment techniques for small area case study’. Overall, it is a comprehensive article and the findings provided indicate that a great deal of effort was put in. Suggestions for improvements that could be performed to the manuscript prior to its publication are the following:

Title: There is a typo. Probably the authors wanted to use the word territory instead of terrority. Therefore, it is suggested to use the correct word. Alternatively, the phase ‘The case study of Poland’ could be used.

Abstract:

-It is suggested to avoid acronyms and parentheses in this part of the manuscript. Try to keep the Abstract simple yet informative and focus on the research questions that the authors try to answer through the paper, as well as the novelties/results that this paper presents.

Introduction:

-It would be interesting to add more information (and corresponding references) about the situation in Europe regarding drought phenomena and resume what has been done so far in this field. In addition, it is suggested to add some more comments regarding the novelties that this paper presents. What exact research questions do the authors try to answer? What research gabs does this paper try to fill up regarding to previous similar studies?

Data and case study localization:

-Line 87: Maybe it should be Fig. 1 instead of Fig 2. It is suggested to include a figure in order to present the area of interest and try to keep a sequence regarding the figures’ numbers. For example, figure 4 should be presented in the beginning.

-Line 109: The analysis of the image is not very good and some details (e.g.  the 69 wells that were selected – line 134) cannot be noticed.

-Line 134: Where does the parenthesis [44,81–89] refer to? Is this a reference?

Methods:

-Line 142: What are the remote-based observations that were used and mentioned? It is suggested to elaborate more on this. It is suggested to add more information regarding the remote sensing data that were used (maybe in the previous section).

-Fig. 2- Flowchart: Is it CGDI or GGDI (line 189)?

Discussion:

-Line 228: Typo – ‘in a figure 5’ -> ‘in figure 5’

-What methods and exact workflows could be used in order to combine the results of all these indices that were calculated? It is suggested to add a general workflow regarding this topic.

Conclusions:

- It would be interesting to comment on the following: What could be some possible restrictions or challenges if the proposed methods were implemented in a different area of interest/different case study? Are there any other factors that should be taken into account?

- It is also suggested to underline the novelties that the proposed method offers and to elaborate more on the future work paths.

Comments on the Quality of English Language

Minor editing of English language is required.

Author Response

Dear Reviewer,

Thank you so much for your  effort, for your hard work and suggestions that will improve my paper. I am very grateful for all the help and thorough analysis of my work. Below I would like to respond to the suggested changes and improvements. 

Best regards, Monika BiryÅ‚o 

Title: There is a typo. Probably the authors wanted to use the word territory instead of terrority. Therefore, it is suggested to use the correct word. Alternatively, the phase ‘The case study of Poland’ could be used.

Corrected, I added: (The case study of Poland).

 Abstract:

-It is suggested to avoid acronyms and parentheses in this part of the manuscript. Try to keep the Abstract simple yet informative and focus on the research questions that the authors try to answer through the paper, as well as the novelties/results that this paper presents.

 Corrected, I added: “Based on the research it was concluded that the CCDI, GGDI, WSDI and MSDI indicators can be a useful tool, based on which I could noticed periods of drought, which were not groundwater related drought, but due to low rainfall and snowfall.” And deleted all acronyms.

Introduction:

-It would be interesting to add more information (and corresponding references) about the situation in Europe regarding drought phenomena and resume what has been done so far in this field. In addition, it is suggested to add some more comments regarding the novelties that this paper presents. What exact research questions do the authors try to answer? What research gabs does this paper try to fill up regarding to previous similar studies?

 Corrected, I added: “Central Europe, as it is an area where extreme phenomena do not occur (catastrophic droughts like in southern Europe, hurricanes, monsoons, earthquakes) is not an area of great interest in scientific publications. However, this does not mean that such areas should not be explored. In light of the changing climate, each area should be monitored regularly. The work, based on the example of Poland and its catchment area, attempts to answer the questions whether the area is at risk of drought, how quickly climate change is progressing and what are the causes of changes in water conditions. Similar topics regarding groundwater changes in this part of Europe have been previously discussed among others in publications Krogulec et al., 2020 but only for Warsaw urban area using one piezometer and two wells; Boczon et al, 2020 for area of the Lebiedzianka river basin; Sliwinska et al, 2019 for the same area but without taking into account meteorological indicators. The topic of meteorological indices was discussed in Okoniewska et al., 2021 but concerning indices as UTCI, STI, Oh_H, WL, and OV; in Kalbarczyk and Kalbarczyk, 2022 and Wicher-Dysarz et al., 2022 concerning Standardized Precipitation Index”.

Data and case study localization:

-Line 87: Maybe it should be Fig. 1 instead of Fig 2. It is suggested to include a figure in order to present the area of interest and try to keep a sequence regarding the figures’ numbers. For example, figure 4 should be presented in the beginning.

Corrected, figure 1 is now figure 2, figure 4 is now figure 1.

-Line 109: The analysis of the image is not very good and some details (e.g.  the 69 wells that were selected – line 134) cannot be noticed.

I wanted to show in this figure exactly how many wells there are in Poland. But since not all of them have a complete time history, I removed some of them for research. I deleted the reference to Figure 2 in the text (now it is Figure 1), because it may actually be misleading, as the drawing shows many more wells than 69. And the figure is now in a different chapter, I think this deletion should not be a problem.

-Line 134: Where does the parenthesis [44,81–89] refer to? Is this a reference?

 Yes, this is a reference to the Bulletines of Sadurski; in the reference list 17-27. In the final version will be addressed to numbers from a reference list, I rebuilt the reference list and missed this in the text. Thank you for noticing this. I put now: Sadurski 2006-2017.

Methods:

-Line 142: What are the remote-based observations that were used and mentioned? It is suggested to elaborate more on this. It is suggested to add more information regarding the remote sensing data that were used (maybe in the previous section).

In the text was mentioned: “remote-based observations from the GRACE mission”. GRACE mission, which is remote sensing based, was described in chapter 2. I added here: “in a form of total water storage changes”, maybe it will make this part clearer.

-Fig. 2- Flowchart: Is it CGDI or GGDI (line 189)?

 Yes, it is GDI, it is changed in the flowchart

Discussion:

-Line 228: Typo – ‘in a figure 5’ -> ‘in figure 5’

Corrected

-What methods and exact workflows could be used in order to combine the results of all these indices that were calculated? It is suggested to add a general workflow regarding this topic.

Thank you very much for this attention. Wouldn't the flowchart in Figure 3 be sufficient? Because it is also related to the data used. I thought that adding an additional flowchart would cause confusion, and she also had no intention of combining the studies into one large one. I treated all these indices as separate parameters which, after analysis, will give a general picture of water conditions

Conclusions:

- It would be interesting to comment on the following: What could be some possible restrictions or challenges if the proposed methods were implemented in a different area of interest/different case study? Are there any other factors that should be taken into account?

Corrected. I added a conclusion: “The proposed methods for determining water indices can be used in almost any region. And I think it would be worth implementing them in continuous monitoring of basin areas. What is particularly important, in the case of testing the resources and availability of groundwater, which is crucial for consumption. However, the porosity coefficient should not be used in the case of areas covered with ice, because the ice itself has a significant impact on the permeability there and the ice itself could be treated as a rock, which is only an additional, important factor.”

- It is also suggested to underline the novelties that the proposed method offers and to elaborate more on the future work paths.

Corrected. I added in the research: “The research on the mentioned indices is important and has not been discussed before in the areas presented in the publication. As mentioned, the studies were conducted for much smaller areas using direct measurements or using a small range of indices

Reviewer 2 Report

Comments and Suggestions for Authors

Minor Reviews:

1. Some sentences are quite lengthy and complex. Consider breaking them down into smaller sentences for better readability.

2. Ensure consistent formatting for in-text citations. Check for missing punctuation or formatting issues.

Consider explaining acronyms when they are first introduced to ensure that readers understand them. For example, TWS, GRACE, GDI, TWSA, etc.

3.  Avoid repetition of certain phrases, such as "extremely important." Consider using synonyms or rephrasing to enhance clarity.

4.  Correct the heading 4.3

5.  Ensure that sentence structures are consistent throughout the document. Some sentences could be restructured for improved flow.

Major Reviews:

1.     Novelty of work is missing.

2.     The introduction is somewhat brief. Consider providing a more comprehensive introduction to drought as a phenomenon and its global significance before delving into specific studies and indices. This will provide context for readers who may not be familiar with the topic.

3.     Consider adding a concluding section that summarizes the key findings and insights from the studies mentioned. This will tie the document together and help readers grasp the broader implications of the research.

4.     Ensure that all references are correctly formatted and cited. Check for consistency in citation style (e.g., APA, MLA, etc.) throughout the document.

5.     Interpret the results of the analysis. What do the values and trends signify? Do they support or challenge existing theories or hypotheses?

6.     Ensure that units of measurement are consistent throughout the document. In some places, you mention values in centimeters, while in others, it's not specified.

7.     Discuss the significance of the observed time lag in the data, especially in cases where it is mentioned (e.g., the two-month time lag between basins).

8.     Review the document for grammatical errors and improve the clarity of language. Ensure that each sentence is concise and easy to understand.

9.     Go beyond stating that droughts have occurred and discuss the potential impacts of these droughts on the environment, society, or the economy of the region.

10.  Summarize the key takeaways from the research and conclude the document with a clear statement about the significance of the findings.

Comments on the Quality of English Language

Moderate editing of English language required

Author Response

Dear Reviewer,

Thank you so much for your  effort, for your hard work and suggestions that will improve my paper. I am very grateful for all the help and thorough analysis of my work. Below I would like to respond to the suggested changes and improvements. 

Best regards, Monika BiryÅ‚o 

Some sentences are quite lengthy and complex. Consider breaking them down into smaller sentences for better readability. I did my best Ensure consistent formatting for in-text citations. Check for missing punctuation or formatting issues.Consider explaining acronyms when they are first introduced to ensure that readers understand them. For example, TWS, GRACE, GDI, TWSA, etc. Corrected Avoid repetition of certain phrases, such as "extremely important." Consider using synonyms or rephrasing to enhance clarity. Corrected Correct the heading 4.3Corrected Ensure that sentence structures are consistent throughout the document. Some sentences could be restructured for improved flow. I did my best  Novelty of work is missing

I added: “Central Europe, as it is an area where extreme phenomena do not occur (catastrophic droughts like in southern Europe, hurricanes, monsoons, earthquakes) is not an area of ​​great interest in scientific publications. However, this does not mean that such areas should not be explored. In light of the changing climate, each area should be monitored regularly. The work, based on the example of Poland and its catchment area, attempts to answer the questions whether the area is at risk of drought, how quickly climate change is progressing and what are the causes of changes in water conditions. Similar topics regarding groundwater changes in this part of Europe have been previously discussed among others in publications Krogulec et al., 2020 but only for Warsaw urban area using one piezometer and two wells; Boczon et al, 2020 for area of the Lebiedzianka river basin; Sliwinska et al, 2019 for the same area but without taking into account meteorological indicators. The topic of meteorological indices was discussed in Okoniewska et al., 2021 but concerning indices as UTCI, STI, Oh_H, WL, and OV; in Kalbarczyk and Kalbarczyk, 2022 and Wicher-Dysarz et al., 2022 concerning Standardized Precipitation Index.”

 The introduction is somewhat brief. Consider providing a more comprehensive introduction to drought as a phenomenon and its global significance before delving into specific studies and indices. This will provide context for readers who may not be familiar with the topic. I added: “Due to direct influence of droughts on people and businesses there is a need of continuous assessment, observation and prevention of this phenomena, as it can be a cause extreme weather phenomena all over the planet. Drought can occur in areas with high and low rainfall because it results from the balance between precipitation and evapotranspiration.  Drought intensity is a relative factor. It seems to depend on its duration, intensity and scope of the drought episode. However, requirements would also need to be taken into account caused by human activity and vegetation. Even short-term droughts can impact society for many years.”  Consider adding a concluding section that summarizes the key findings and insights from the studies mentioned. This will tie the document together and help readers grasp the broader implications of the research.  Ensure that all references are correctly formatted and cited. Check for consistency in citation style (e.g., APA, MLA, etc.) throughout the document. Corrected  Interpret the results of the analysis. What do the values and trends signify? Do they support or challenge existing theories or hypotheses?  Ensure that units of measurement are consistent throughout the document. In some places, you mention values in centimeters, while in others, it's not specified.Discuss the significance of the observed time lag in the data, especially in cases where it is mentioned (e.g., the two-month time lag between basins). I checked all the paper, units are now consistent. I added an explanation: “The lag shows that in recent years, in the Odra basin area, the maximum and minimum CCDI levels are reached two months earlier than in the case of the Vistula River. This proves a higher ratio of precipitation to evaporation in the Odra basin area. This is understandable, because the western part of Poland has a warmer climate and a much earlier spring, as well as a shorter winter.”  Review the document for grammatical errors and improve the clarity of language. Ensure that each sentence is concise and easy to understand. I did my best Go beyond stating that droughts have occurred and discuss the potential impacts of these droughts on the environment, society, or the economy of the region.  Corrected. I added: “Long-term droughts may cause land subsidence, which poses a threat to the stability of the ground, foundations and streets. Huge damage can also be seen in ecosystems. Some areas may even become unusable, and some will have to completely change crops to less demanding plants. Groundwater shortages lead to shortages of drinking water, which leads to local governments' decisions to limit water consumption to watering lawns or irrigating home crops. And such decisions lead to smaller crops and an increase in the costs of purchasing them” Summarize the key takeaways from the research and conclude the document with a clear statement about the significance of the findings. Corrected. I added: “The proposed methods for determining water indices can be used in almost any region. And I think it would be worth implementing them in continuous monitoring of basin areas. What is particularly important, in the case of testing the resources and availability of groundwater, which is crucial for consumption. However, the porosity coefficient should not be used in the case of areas covered with ice, because the ice itself has a significant impact on the permeability there and the ice itself could be treated as a rock, which is only an additional, important factor.”

Reviewer 3 Report

Comments and Suggestions for Authors

Dear Editor,

Thank you for inviting me as a reviewer for manuscript “Remote sensing based hydro-extremes assessment techniques for small area case study (Polish terrority)”. The article is presents an interesting topic. This work is meaningful and the result is basically satisfactory. However, some other problems in the manuscript are still concerned in the following:

Major comments 

  1. The abstract is very general.  Authors should avoid general descriptions in the abstract and should include more results. Please also describe the research method and data used based on the structure of the abstract.
  2. The paper lacks to describe related works as well as a state of the art of research including the methods used in literature making not clear the originality and soundness of the research work. 
  3. Authors should enhance the introduction section by adding more recent and relevant articles related. The authors need to discuss their contributions compared to those in related papers. From that, the discussion author will provide a proper research question and the contribution of the paper. The contribution of the paper should be clearly stated. 
  4. All figures should change to a high quality and clear status. 
  5. The authors did not provide sufficient details on the data used. More information should be provided about the limitations of use these data. 
  6. Standard tables can be used to describe the data of satellite products, rather than dwelling on describing them. 
  7. In the section Discussion, do include the relevance of your work and explain why it is important, i.e., its relevance and benefit. Compare the results presented in the manuscript with results presented in relevant papers published earlier by other authors. 
  8. Conclusions need to be rewritten to highlight the novelty and/or findings of this study. 

Best regards 

Author Response

Dear Reviewer,

Thank you so much for your  effort, for your hard work and suggestions that will improve my paper. I am very grateful for all the help and thorough analysis of my work. Below I would like to respond to the suggested changes and improvements. 

Best regards, Monika BiryÅ‚o 

The abstract is very general.  Authors should avoid general descriptions in the abstract and should include more results. Please also describe the research method and data used based on the structure of the abstract.

Corrected. I changed an abstract a bit, moreover I added some results to abstract: “Based on the research it was concluded that the CCDI, GGDI, WSDI and MSDI indicators can be a useful tool, based on which I could noticed periods of drought, which were not groundwater related drought, but due to low rainfall and snowfall.”

The paper lacks to describe related works as well as a state of the art of research including the methods used in literature making not clear the originality and soundness of the research work. 

Corrected. I added to the beginning of introduction: “Due to direct influence of droughts on people and businesses there is a need of continuous assessment, observation and prevention of this phenomena, as it can be a cause extreme weather phenomena all over the planet. Drought can occur in areas with high and low rainfall because it results from the balance between precipitation and evapotranspiration.  Drought intensity is a relative factor. It seems to depend on its duration, intensity and scope of the drought episode. However, requirements would also need to be taken into account caused by human activity and vegetation. Even short-term droughts can impact society for many years [Wilhite and Glantz, 1985].”And also at the end of introduction part: “Central Europe, as it is an area where extreme phenomena do not occur (catastrophic droughts like in southern Europe, hurricanes, monsoons, earthquakes) is not an area of great interest in scientific publications. However, this does not mean that such areas should not be explored. In light of the changing climate, each area should be monitored regularly. The work, based on the example of Poland and its catchment area, attempts to answer the questions whether the area is at risk of drought, how quickly climate change is progressing and what are the causes of changes in water conditions. Similar topics regarding groundwater changes in this part of Europe have been previously discussed among others in publications Krogulec et al., 2020 but only for Warsaw urban area using one piezometer and two wells; Boczon et al, 2020 for area of the Lebiedzianka river basin; Sliwinska et al, 2019 for the same area but without taking into account meteorological indicators. The topic of meteorological indices was discussed in Okoniewska et al., 2021 but concerning indices as UTCI, STI, Oh_H, WL, and OV; in Kalbarczyk and Kalbarczyk, 2022 and Wicher-Dysarz et al., 2022 concerning Standardized Precipitation Index.”

Authors should enhance the introduction section by adding more recent and relevant articles related. The authors need to discuss their contributions compared to those in related papers. From that, the discussion author will provide a proper research question and the contribution of the paper. The contribution of the paper should be clearly stated. 

Corrected, added at the end of introduction.

All figures should change to a high quality and clear status. 

Corrected.

The authors did not provide sufficient details on the data used. More information should be provided about the limitations of use these data. 

Corrected. I added: “GRACE and MERRA satellite observations are a suitable tool to quickly and cheaply monitor water conditions. An aspect that is both a disadvantage and an advantage - the existence of an unprecedented possibility of global surveys and surveys of large areas, however, this resolution limits the possibilities of assessing small areas and makes point surveys impossible.”

Standard tables can be used to describe the data of satellite products, rather than dwelling on describing them. 

I shorten  describtions.

In the section Discussion, do include the relevance of your work and explain why it is important, i.e., its relevance and benefit. Compare the results presented in the manuscript with results presented in relevant papers published earlier by other authors. 

In the first version of the paper, I started this part with: “Standardized indices can become a very helpful tool in hydrological and climatic research. Their advantage is that they combine plenty of variables that have an influence on climatic changes.” I think here I perform the relevance of my work. I added also: “Thorough and constant monitoring of water levels carried out through designated indices would allow for a faster response to changing levels and prevent or remedy droughts or floods. Such an analysis will also allow for better planning in agriculture, water management and rational use of drinking groundwater.”

As I wrote in the paper, such research was not performed for the region before. That is why I am not able to compare it with similar research.

Conclusions need to be rewritten to highlight the novelty and/or findings of this study. 

Corrected. I added: “The research on the mentioned indices is important and has not been discussed before in the areas presented in the publication. As mentioned, the studies were conducted for much smaller areas using direct measurements or using a small range of indices”

Round 2

Reviewer 1 Report

Comments and Suggestions for Authors

The proposed suggestions for improvement were implemented.

Comments on the Quality of English Language

Minor editing of English language is required.

Author Response

Dear Reviewer, thank you for all your help, support and contrubution.

Best regards, Monika

Reviewer 3 Report

Comments and Suggestions for Authors

Dear Editor,

Thank you again for inviting me to review this exciting manuscript. The authors responded to my concerns well. Accordingly I recommend acceptance of the manuscript.

Best regards

Author Response

(The authors gave the same response as above.)
